# Building Sub-Saharan African PBPK Populations Reveals Critical Data Gaps: A Case Study on Aflatoxin B1

**DOI:** 10.3390/toxins17100493

**Published:** 2025-10-03

**Authors:** Orphélie Lootens, Marthe De Boevre, Sarah De Saeger, Jan Van Bocxlaer, An Vermeulen

**Affiliations:** 1Centre of Excellence in Mycotoxicology and Public Health, Department of Bioanalysis, Ghent University, 9000 Ghent, Belgium; 2Laboratory of Medical Biochemistry and Clinical Analysis, Department of Bioanalysis, Ghent University, 9000 Ghent, Belgium; 3MYTOX-SOUTH^®^, International Thematic Network, 9000 Ghent, Belgium; 4Cancer Research Institute Ghent (CRIG), 9000 Ghent, Belgium; 5Department of Biotechnology and Food Technology, University of Johannesburg, Johannesburg 2028, South Africa

**Keywords:** CYP450-enzymes, pharmacokinetics, genetics, PBPK, diversity, Sub-Saharan Africa

## Abstract

Physiologically based pharmacokinetic (PBPK) models allow to simulate the behaviour of compounds in diverse physiological populations. However, the categorization of individuals into distinct populations raises questions regarding the classification criteria. In previous research, simulations of the pharmacokinetics of the mycotoxin aflatoxin B1 (AFB1), were performed in the black South African population, using PBPK modeling. This study investigates the prevalence of clinical CYP450 phenotypes (CYP2B6, CYP2C9, CYP2C19, CYP2D6, CYP3A4/5) across Sub-Saharan Africa (SSA), to determine the feasibility of defining SSA as a single population. SSA was subdivided into Central, East, South and West Africa. The phenotype data were assigned to the different regions and a fifth SSA group was composed of all regions’ weighted means. Available data from literature only covered 7.30% of Central, 56.9% of East, 38.9% of South and 62.9% of West Africa, clearly indicating critical data gaps. A pairwise proportion test was performed between the regions on enzyme phenotype data. When achieving statistical significance (*p* < 0.05), a Cohen’s d-test was performed to determine the degree of the difference. Next, per region populations were built using SimCYP starting from the available SSA based *SouthAfrican_Population FW_Custom* population, supplemented with the phenotype data from literature. Simulations were performed using CYP probe substrates in all populations, and derived PK parameters (C_max_, T_max_, AUC_ss_ and CL) were plotted in bar charts. Significant differences between the African regions regarding CYP450 phenotype frequencies were shown for CYP2B6, CYP2C19 and CYP2D6. Limited regional data challenge the representation of SSA populations in these models. The scarce availability of in vivo data for SSA regions restricted the ability to fully validate the developed PBPK populations. However, observed literature data from specific SSA regions provided partial validation, indicating that SSA populations should ideally be modelled at a regional level rather than as a single entity. The findings, emerging from the initial AFB1-focused PBPK work, underscore the need for more extensive and region-specific data to enhance model accuracy and predictive value across SSA.

## 1. Introduction

Physiologically based pharmacokinetic (PBPK) modeling is widely used in pharmacology, providing a robust framework for a mechanistic understanding of the dynamics of drug absorption, distribution, metabolism, and excretion (ADME) within biological systems. Population-based PBPK models incorporate variability and covariates to reflect a more realistic and comprehensive frame of how different individuals within a population handle a drug. The development and refinement of population-based PBPK models has emerged as a pivotal tool in drug research and development, interaction studies, clinical practice, and regulatory decision-making and has evolved substantially over the past few decades, driven by enhanced physiological understanding and advanced computational methodologies. PBPK modelling has progressively accommodated the inherent variabilities within populations, thereby enabling more precise predictions of drug PK across diverse populations and representing human diversity [1,2].

Although most applications of PBPK modeling have focused on drugs, its use has over the years been extended to environmental contaminants (such as persistent organic pollutants, melamine, seafood toxins, food contaminants a.o.) and mycotoxins, providing insight into exposure, metabolism and risk assessment [3,4,5,6,7,8].

Besides a robust PBPK model, there is the need for the accurate representation of genetic and physiological factors within a population. Genetic differences in enzyme and transporter expression and functionality can lead to significant variability in drug exposure. PBPK models can simulate the impact of genetic polymorphisms on drug PK by incorporating the genotype and phenotype differences in drug metabolizing enzymes and transporters [9]. It is well known that a wide range of genetic polymorphisms can have a potential impact on the PK, more specifically on drug metabolism and transport as well as on drug response or pharmacodynamics (PD) [10,11]. Currently, populations in PBPK softwares are based on geographical (continental) borders rather than on genetic differences. Consequently, there are substantial challenges associated with the building of specific populations such as the African population, since continental geography is a poor substitute for ethnic, thus potentially genetic, partitioning. Moreover, only limited data on these physiological and genetic differences are available [2].

The African continent consists of 54 countries, spread over 30.27 million km^2^ with a population of approximately 1.5 billion people. After Asia, it is the most populous continent in the world [12]. The African continent contains remarkable genetic diversity. African genomes exhibit heterogeneity, resulting from distinct evolutionary trajectories [13]. Each African genome comprises around 25% more polymorphisms compared to non-African genomes. It has been shown that the African continent harbors rare allele variants and that some variants are exclusively found in Africa [14,15,16]. For PBPK modeling this extensive genetic variability has direct implications: polymorphisms affecting drug and other xenobiotic metabolizing enzymes such as CYP450s can lead to wide interregional differences in pharmacokinetics. Understanding the genetic variation that is present in this population is crucial to define additional PBPK populations. Currently, PBPK models primarily rely on default populations, all too often based on healthy adult populations [17]. There are already various populations available in commercial PBPK softwares such as a renal impairment population, a cancer population, as well as age-related populations e.g., pediatric populations and elderly populations. However, considering the wide genomic variability that is harbored on our planet, a better subdivision into relevant populations should be applied. The subdivision of populations based on continental borders does not seem sufficient since this does not account for the wide genomic variations across continents. In the framework of simulating aflatoxin B1 (AFB1) exposure in Sub-Saharan Africa (SSA), based on exposure through contaminated food, a more in-depth look was given to the African continent, more specifically SSA, since the African continent is usually considered as the region above the Sahara desert, Northern Africa, and the region under the desert, SSA [18]. A subdivision of SSA was proposed aligned with the divisions established by the African Union [19,20].

## 2. Results

Guidelines of the European Medicines Agency (EMA) and the Food and Drug Administration (FDA) provided information on the qualification of populations within PBPK models. The process involves the establishment of system-dependent parameters, succeeded by the prospective prediction of PK parameters within the target population for verification. Both EMA and FDA state that the qualification/verification of the population model should be presented and justified for its intended purpose [21,22]. The development of a virtual population necessitates an assessment of the ability of the PBPK platform to generate virtual individuals with physiological, anatomical, biological and genetic values that are similar to observed, in vivo data in this population. Therefore, an evaluation should be made whether a specific population exhibits similarities in body weight, height, age, and tissue volumes compared to observed data. Public health databases serve as a resource for gathering population-related data. Populations can be built de novo or can be developed by changing physiological parameters, demographics, genetic polymorphisms etc. in an already verified population. As mentioned by Shebley et al. in 2018, qualification of the PBPK platform is emphasized by EMA while the FDA focuses on the content of PBPK analyses, including reporting and formatting [23]. The initial literature search in Embase, Medline and Pubmed resulted in 97 articles; a subsequent refinement reduced the number to 19 results.

In Central Africa, only Burundi was included, accounting for approximately 7.30% of the regional population. In Eastern Africa, Ethiopia, Tanzania, and Uganda were represented, covering about 56.9% of the population. In Southern Africa, South Africa and Zimbabwe together represented 38.9%. In Western Africa, Nigeria, Niger, Benin, and Togo were included, representing 62.9% of the regional population. These comparisons indicate that population coverage was higher in Eastern, Southern, and Western Africa, whereas Central Africa remained largely underrepresented, with over 90% of its population not captured. A summarizing table (Table 1) is listed below.

Phenotypic data were found for different Sub-Saharan countries on CYP2B6, CYP2C9, CYP2C19, CYP2D6 and CYP3A4/5 and are reported in Table 2.

The used poor metabolizer (PM) and extensive metabolizer (EM) frequencies per African region are listed in Table 3 and were based on the literature data presented in Table 2.

The statistical testing revealed significant differences for CYP2B6 between SSA and West (*p* = 7.2 × 10^−11^), SSA and East (*p* < 2 × 10^−16^), West and East (*p* = 0.0061), West and South (*p* = 6.9 × 10^−11^) and South and East (*p* < 2 × 10^−16^). For CYP2C19 there was a statistically significant difference between West and East (*p* = 0.0075). For CYP2D6 a statistically significant difference was observed for SSA and Central (*p* = 5.7 × 10^−5^), East and Central (*p* = 9.1 × 10^−5^) and for South and Central (*p* = 2.8 × 10^−6^). For CYP2C9 and CYP3A5, no significant phenotype differences were observed across the regions. Standardized mean differences were reported by calculating Cohen’s d-values for the significant results. The Cohen’s d-values were between 0.4 and 0.5, indicative of a moderate difference only. The prediction bar charts plotted for C_max_, T_max_, AUC and CL by simulating PK parameters for CYP450 probe substrates in the different populations using SimCYP are presented in Appendix A. Notably, variations in C_max_, AUC, and CL were observed for CYP2C9 and CYP3A5 substrates, while differences in C_max_, T_max_, AUC, and CL were observed for CYP2D6 substrates. These Appendix A provide a visual representation of regional variability.

Generally, the African continent is partitioned into two regions, namely Northern Africa and SSA, for population categorization. Numerous studies have substantiated the presence of inter-ethnic diversity within these regions [18,44]. Verification (within twofold) of the built populations was performed by comparing predicted PK parameters to available observed human PK parameters data (Table 4).

In the observations from the CENTRAL region with dihydroartemisinin, CYP2B6 and CYP3A4 are involved [49]. For the observed data for EAST Africa, no CYP enzymes are involved in the metabolism of LAM [46]. For the SOUTH Africa data, more CYP450 enzymes are involved since LPV is a CYP3A4 substrate and RTV is a substrate of CYP3A4, an inhibitor of both CYP2D6 and CYP3A4 and an inducer of CYP2B6 and CYP2C9 [47]. For the observed data for WEST Africa, CYP3A-enzymes are involved in the metabolism of nifedipine [48].

## 3. Discussion

As the PBPK substrate file for AFB1 is primarily driven by CYP450-mediated metabolism, in particular CYP1A2 and CYP3A4, only CYP450 phenotype data were collected, providing the most physiologically relevant basis for interpopulation comparisons. Other enzyme and transporter frequencies could be incorporated in future studies for contaminants with different metabolic pathways. Furthermore, there is a lack of CYP1A2 and CYP3A4 phenotype and allele frequency data in SSA populations relevant to AFB1 metabolism, limiting the ability to perform detailed inter-regional comparisons. For PBPK modeling of AFB1 metabolism, phenotype data remain essential to accurately capture enzyme activity and interindividual variability. The scarcity of CYP1A2 and CYP3A4 phenotype and allele frequency data in SSA populations significantly limits the predictive accuracy of PBPK population models for AFB1. Furthermore, CYP3A4 is a major enzyme involved in the metabolism of a wide range of xenobiotics, including drugs and thus is crucial for the population model. Future studies should focus on phenotyping and could complement phenotyping with genotype- or haplotype-based analyses to provide broader population coverage and improve regional extrapolations [50,51,52].

Due to a lack of sufficient data, CYP450 phenotype data were compiled into PM and EM. Reducing CYP450 phenotypes to only PM and EM categories, however, fails to capture the full spectrum of metabolic variability (PM, intermediate metabolizers (IM), EM and ultra-rapid metabolizers (UM)). The current data collection focused on phenotypes and not on genotypes since their relationship is not always straightforward. Some individuals may have a genotype predicting an EM status, though phenotypically they present as IMs or even PMs [53]. The latter can be explained by the impact of multiple influencing factors such as e.g., environmental factors, that influence phenotype expression [54]. Although the focus was on phenotype rather than on genotype data, relying solely on a PM versus EM classification may lead to suboptimal dosing strategies, potentially resulting in underdosing in UMs and overdosing in IMs. This was a first drawback in developing a PBPK population model having only limited data available [50,51,52].

The incomplete coverage of data is clearly illustrated in Figure 1, which shows the countries within each SSA region from which CYP450 phenotype data could be retrieved.

This lack of comprehensive data resulted in regional models that were not fully representative of the entire subcontinent, as previously shown in the data of Table 1. Consequently, the regional subdivision became less accurate, raising the question if population models should rather have been developed at a country-specific level (e.g., South Africa, Tanzania, Zimbabwe) to provide a more precise and reliable understanding of the genetic diversity in each area.

The significant differences in CYP450 enzyme phenotype frequencies (for CYP2B6, CYP2C19 and CYP2D6) across SSA regions impact the development and application of PBPK models, which predict ADME of drugs and the outcome of interaction studies in various populations [55,56,57]. Concerning the impact on the PK of AFB1, these differences will not have an influence since AFB1 is metabolized via CYP1A2 and CYP3A4 in humans, and data on these CYP450 enzymes were missing [8]. The differences for CYP2B6, CYP2C19 and CYP2D6 could have important implications for the metabolism and disposition of drugs that are substrates and/or precipitants of these enzymes. More extensive enzyme phenotype data of SSA countries are necessary to comprehensively evaluate whether other differences might be significant (e.g., CYP3A4/5 enzyme phenotype frequencies across SSA). The accuracy of these models depends on reliable, population specific data on CYP450 enzyme phenotype frequencies and other physiological parameters. Incorporating data on genetic polymorphisms and on CYP450 enzyme abundancies could enhance the predictive accuracy of this model, however, literature revealed an insufficient quantity of relevant data from the four SSA regions to utilize as Appendix A.

The incomplete evaluation of CYP450 phenotype data in certain regions is primarily due to a lack of available information. This poses a significant limitation to the study, as it leaves gaps in understanding how these enzymes affect drug metabolism in different populations. However, the authors chose to publish this manuscript to raise awareness about this data gap and to encourage further research in these unverified areas, recognizing the need for more comprehensive studies in the future. Furthermore, the study aimed to divide SSA into four regions and collect data on CYP450 phenotypes. However, the data collection process led to limitations, as the available data did not cover all countries within each region and phenotype data had to be compiled into PM and EM groups only [19,20].

The observed differences in CYP2B6 frequencies between SSA and West, East and South Africa suggest that the metabolism of CYP2B6 substrates may vary significantly across these populations [58]. Concerning CYP3A4/5, literature data was only available for South. For Central, East and West the available data from the SimCYP population were used. The SSA population consisted of the weighted mean of the literature data, meaning that the same phenotype frequency as the South region were used. Therefore, no conclusion can be made on differences between the regions for CYP3A4/5. Data on this CYP phenotype are crucial since differences in CYP3A4/5 frequencies could impact the PK of drugs like LPV and RTV, which are metabolized by this enzyme and which are commonly used in combination therapy for HIV/AIDS [59]. Upon examination of Table 2, it is evident that there are variations within the selected regions (e.g., PM frequency for CYP2B6 for Zimbabwe being 0.14 and 0.734 in two different studies) [28,32]. These discrepancies may be attributed to genuine phenotypic differences within subregions of each region, for example in Matimba et al., 2008 the phenotype data are clustered from the Shona, San and mixed group inhabitants [28]. Alternatively, the observed differences might result from an insufficient amount of phenotypic data, which is necessary to adequately represent the region. Also the clustering of PM and IM and of EM and UM, which was performed because of a lack of data, might be contributing to the differences. The CYP450 phenotype frequencies derived from the literature in this research differ from those used in the *SouthAfrican_Population FW_Custom*, as shown in Table 3. Notably, the frequency for CYP3A4/3A5 PM is 0.726 according this literature review, compared to 0.180 in the existing SimCYP population. As previously mentioned, due to a lack of data on CYP3A4/3A5 phenotype frequencies, the literature data from Zimbabwe was used to represent SSA. The SimCYP population was built on CYP450 phenotype frequency data from different, multiple African countries. The development of the Southern African population model in SimCYP, starting from the Sim-NEur Caucasian population by incorporating region-specific adjustments based on available demographic and genetic data, is an efficient way to develop population-specific PBPK models.

In vivo data on the newly described populations were, however, scarce [45,46,47,48]. For Central Africa no data in healthy, adult populations were found in literature and therefore data in a pediatric malarian population were used for population verification [45]. Model performance varied across regions due to differences in the availability and completeness of CYP450 phenotype data, population demographics, and the limited number of drugs with reported in vivo studies, highlighting critical data gaps in African populations. There is a need for more in vivo data on healthy, adult volunteers in these regions in order to fully verify the developed populations. Based on the sparse available data, the PBPK model could not be verified for West Africa since the predicted T_max_ was not within twofold of the observed T_max_. If more data would be available, the verification of the developed populations would be more reliable.

The moderate effect sizes (Cohen’s d values of 0.4–0.5) indicate that these population differences, while statistically significant, may not necessarily translate to large differences in PK parameters. However, in the context of drugs with narrow therapeutic indices or with a complex metabolic pathway, even moderate differences could be clinically relevant [60]. It could lead to suboptimal dosing, increased risk of toxicity or therapeutic failure. This underscores the importance of considering population specific variability when developing dosing strategies. The results of this study highlight several areas that require further research and investigation.

There is a need for more research on the clinical implications of CYP450 enzyme phenotype frequency differences in SSA populations. This research can help to better understand the impact of these differences on treatment outcomes and inform the development of more effective and safer treatment regimens.

In summary, the findings from this analysis emphasize the need for a thorough understanding of population-specific differences in CYP450 enzymes phenotype frequencies and other physiological parameters when developing PBPK models and conducting PK assessments, particularly in diverse regions, like SSA. This knowledge can inform more accurate predictions of drug bioavailability, metabolism and disposition and guide the optimization of dosing regimens for improved therapeutic outcomes [18,44].

## 4. Conclusions

To date, the development of a PBPK population is not regulated by the EMA and FDA, only guidelines are provided. New populations can be built de novo or based on an already existing default population. In this research, new populations were built based on the *SouthAfrican_Population FW_Custom* population and were supplemented with CYP450 phenotype frequency data. Verification was attempted with in vivo data on model substrates; however, this was constrained by limited availability of appropriate data. While some regions could be reasonably verified (within twofold), validation relied on pediatric data in certain cases and did not fully meet the twofold criterion for West Africa, highlighting that these models remain preliminary. Data on CYP1A2 phenotypes were missing in literature, a crucial CYP450 enzyme for the metabolism of AFB1 in humans. Furthermore, data on CYP3A4 was also scarce, only data from Zimbabwe was found, also compromising the possibility to verify differences between the different African regions concerning CYP3A4, a main metabolizing enzyme for the metabolism of AFB1. The modeling of AFB1 in the African population models remains therefore preliminary. Statistical analyses indicated that for CYP2B6, CYP2C19 and CYP2D6 significant differences are present between certain regions. Numerous studies have substantiated the presence of inter-ethnic diversity within the SSA population [18,44] and this study justifies the subdivision of SSA into Central, East, South and West Africa.

PBPK modeling has been extensively used to predict drug disposition and metabolism, and its application to environmental contaminants, including mycotoxins is emerging. Despite the high prevalence of AFB1 exposure in SSA, few PBPK models have been developed for African populations, largely due to limited physiological, metabolic, and demographic data. This work addresses this gap by attempting to build representative African virtual populations for PBPK simulations of AFB1 exposure. Our findings highlight critical limitations in available data, particularly for CYP450 phenotype frequencies, underscoring the need for comprehensive studies. By systematically documenting these data gaps, this study provides a foundation for future PBPK modeling efforts and demonstrates the importance of generating population specific information to accurately assess AFB1 exposure in African populations.

Further research should be performed to investigate if further subdivisions in these regions should be implemented. In conclusion, this study highlighted the necessity for more comprehensive research on genotypic and phenotypic data within populations to enhance the PBPK population models that are currently applied. Potential strategies to address this data gap include establishing pan-African biobanks and collaborative databases, which could facilitate standardized collection, sharing and integration of population-specific data across the continent.

## 5. Materials and Methods

A literature search was performed to get insight into the guidelines from the EMA and the FDA on how to define PBPK populations. Additionally, a literature review was carried out using Embase, Medline and PubMed to collect phenotype data on enzyme activities for the adult SSA healthy population. Exclusion of articles was made based on the title, abstract and/or full text. Articles were only excluded in the first step if the title explicitly fell out of the reviewing scope. The flowchart on the collection of the phenotype data and the used Boolean strings are shown in Figure 2.

Studies were included if they reported phenotype frequencies for at least one CYP450 enzyme in an African population. Exclusion criteria were studies not conducted in African populations, studies lacking phenotype data, or duplicate datasets. Given the scarcity of available data, no formal risk-of-bias or study quality assessment was applied; instead, all eligible studies were included, and discrepancies were addressed by reporting ranges where applicable. Data extraction focused on CYP450 phenotype frequencies (PM, IM, EM, UM) relevant for population modeling.

Phenotype data for various countries were compiled per CYP450 enzyme (i.e., CYP2B6, CYP2C9, CYP2C19, CYP2D6 and CYP3A4/5) and were classified PM or EM categories [61]. IM frequencies were added to the PM group, while UM data were incorporated into the EM group. Next, 5 populations (SSA, Central, East, South and West) were developed using the SimCYP^®^ software (v21), starting from *the SouthAfrican_Population FW_Custom* population which was modified with the CYP450 frequency data per African region [8]. This custom population file provides a physiologically relevant baseline already parameterized with African characteristics and validated by SimCYP.

Mentionworthy, the *SouthAfrican_Population FW_Custom* population, available in the SimCYP Member Area, was built starting from the existing Sim-NEur Caucasian population. A Weibull function was selected for the age distribution, and the alpha and beta values were selected for both males and females based on the best fit with observed data provided by SimCYP. Different coefficients for the dependence of body weight on height were used for the *SouthAfrican_Population FW_Custom* compared to the Sim-NEur Caucasian population. Regarding the body surface area (BSA), the Nwoye formula was used based on a non-obese, Nigerian male population of 20 subjects.

The regional subdivision of SSA was based on Figure 3, where the entire African continent is divided into 6 regions namely, North, Central, East, South, West and Diaspora [19,20]. For the current study, only CYP450 phenotype data from countries from Central, East, South and West were collected, being part of SSA.

The used PM and EM frequencies per African region were calculated as weighted means based on the used literature data and were incorporated into the *SouthAfrican_Population FW_Custom* provided by SimCYP (v21). The SSA population consists of the available SimCYP population supplemented with the weighted means across all regions. In cases where phenotype frequency data for a specific enzyme were unavailable in a given region, the SSA averages, from the SimCYP custom file, were used to supplement the missing values. This approach allowed regional PBPK populations to be constructed while minimizing bias introduced by incomplete regional data.

The representativeness of the study was evaluated by comparing the number of countries and corresponding populations included with the total populations of the four subregions of SSA (Central, Eastern, Southern, and Western Africa), based on the World Bank population estimates, using the Health Nutrition and Population Statistics Dataset [24].

A summary was made detailing the phenotype data per CYP450 enzyme, including the number of individuals assessed for phenotype frequency determination. Next, using R Studio (v2023.03), a pairwise proportions test with Bonferroni correction was performed on the phenotype frequency data across the 5 regions per CYP450 enzyme. In case of significance, a Cohen’s d-test to verify for standardized mean differences was applied to check the degree of the difference (small, medium or large effect size). Next, simulations in the 5 populations (SSA, Central, East, South and West) with 5 probe substrates were performed using the standard of care once daily dosing in SimCYP, i.e., 600 mg efavirenz (CYP2B6), an HIV-blocker, 10 mg warfarin (CYP2C9), a vitamin K antagonist, 200 mg mephenytoin (CYP2C19), an anti-epilepticum, 22 mg dextromethorphan (CYP2D6), an antitussivum, and 5 mg midazolam (CYP3A4/5), a benzodiazepine These doses were chosen because they correspond to standard therapeutic or widely used clinical probe doses for each CYP enzyme, allowing the simulations to reflect clinically relevant exposures. The population size consisted of 1000 healthy adult volunteers (age: 18–64 y/o; proportion of males: 50%) and the treatment period was 30 days. The lower and upper age limits were set to 18 and 64 y/o, respectively, to exclude the pediatric and elder population and to mimic the Boolean search string that was applied during the literature search. Bar charts were plotted, as an illustration, for each region for the following PK parameters i.e., C_max_, T_max_, AUC_ss_, CL (Appendix A).

The built populations were verified (within twofold) in SimCYP through the prediction of PK parameters, which were then compared with observed PK parameters obtained from literature, containing observed human data [45,46,47,48]. For the Central African population, no available PK data was found in an adult, healthy population. A prediction was performed for C_max_, T_max_ and AUC after an exposure to 4 mg/kg of dihydroartemisinin once daily for three days and compared to the data available from literature in a study conducted in Gabon in pediatric malarian patients [45]. For the East African population, a prediction was performed for C_max_, T_max_ and AUC_ss_ after an exposure to lamivudine (LAM) 150 mg twice daily for one day and compared to the data available from literature in a study conducted in Ugandans [46]. For the South African population, a prediction was performed for C_max_, T_max_ and AUC_ss_ after an exposure to 400 mg lopinavir (LPV) and 100 mg ritonavir (RTV) twice daily for 31 days and compared to observed data from a study in South Africa [47]. For the West African population, a prediction was performed for C_max_, T_max_ and AUC_ss_ after a single exposure to 20 mg of nifedipine and compared to the data available from a study in Nigerians [48].

## Figures and Tables

**Figure 1 toxins-17-00493-f001:**
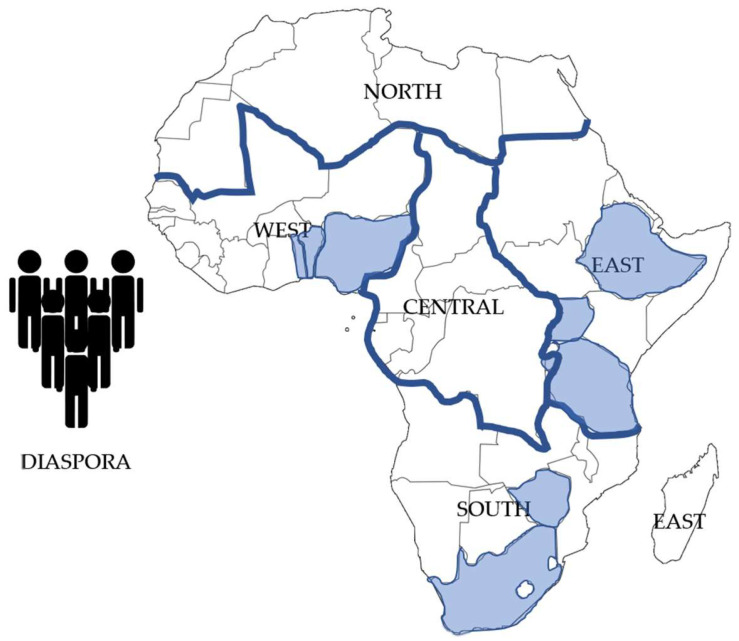
The six regions of the African Union and their constituent countries. Countries from which CYP450 phenotype data could be retrieved are highlighted (blue) with a focus on Central (only data from 1/9 countries), East (only data from 3/14 countries), South (only data from 2/10 countries) and West (only data from 3/15 countries) [19,20].

**Figure 2 toxins-17-00493-f002:**
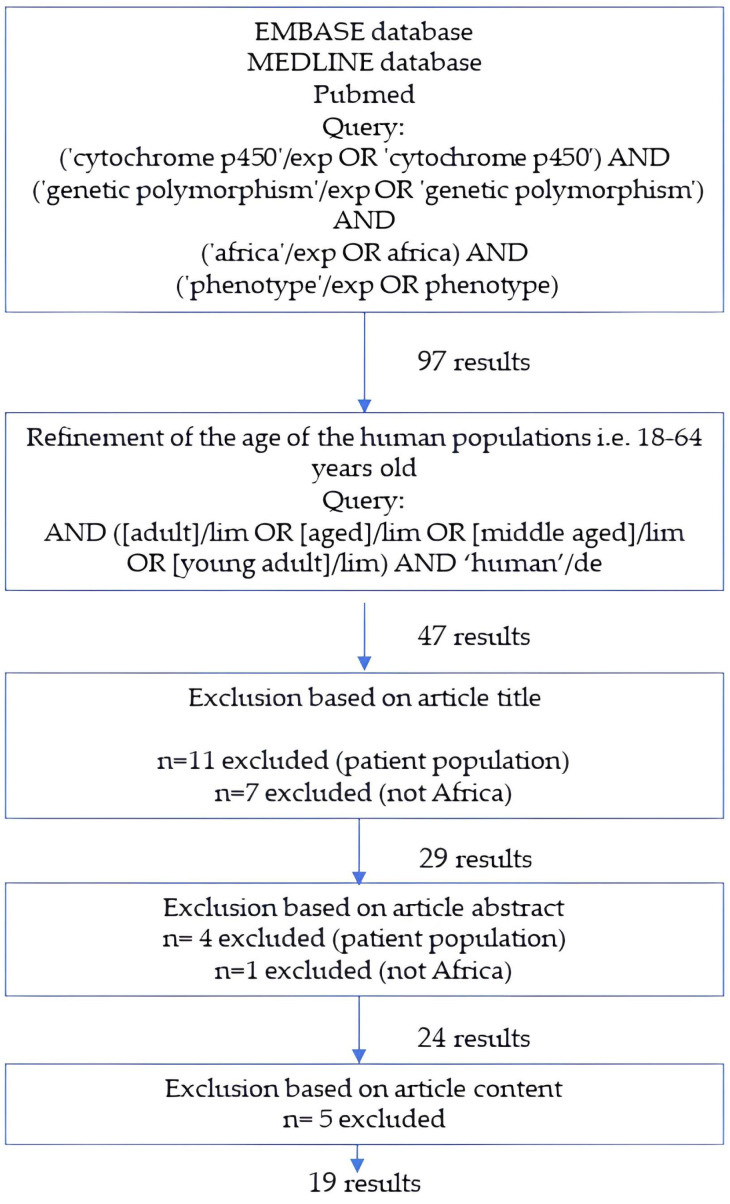
Flowchart of the database search for the collection of phenotype data for CYP450 enzymes in Sub-Saharan Africa (SSA).

**Figure 3 toxins-17-00493-f003:**
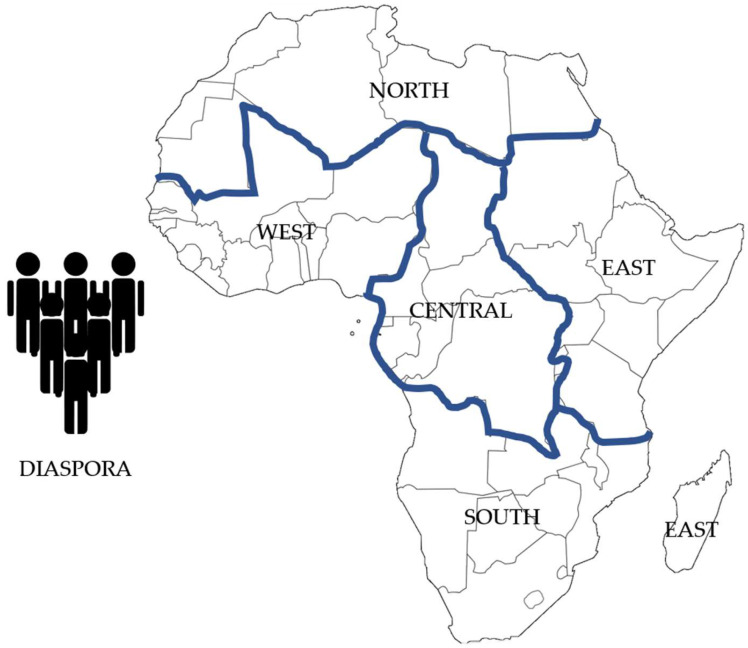
The six regions of the African Union and their constituent countries being North (7 countries: Algeria, Egypt, Libya, Mauritania, Morocco, Sahrawi Arab Democratic Republic and Tunisia), Central (9 countries: Burundi, Cameroon, Central African Republic, Chad, Congo, Democratic Republic of Congo, Equatorial Guinea, Gabon and São Tomé-and-Principe), East (14 countries: Comoros, Djibouti, Ethiopia, Eritrea, Kenya, Madagascar, Mauritius, Rwanda, Seychelles, Somalia, South Sudan, Sudan, Tanzania and Uganda), South (10 countries: Angola, Botswana, Lesotho, Malawi, Mozambique, Namibia, South Africa, Swaziland, Zambia and Zimbabwe), West (15 countries: Benin, Burkina Faso, Cabo Verde, Côte d’Ivoire, Gambia, Ghana, Guinea-Bissau, Guinea, Liberia, Mali, Niger, Nigeria, Senegal, Sierra Leone and Togo), and Diaspora [19,20].

**Table 1 toxins-17-00493-t001:** Representation of the regional coverage of the countries and populations in Sub-Saharan Africa included in the study [24].

Region	Countries Included	Included Population	Total Region Population	% Covered	% Missing
Central	Burundi	13,200,000	180,619,000	7.30	92.7
East	Ethiopia, Tanzania, Uganda	242,500,000	426,371,900	56.9	43.1
South	South Africa, Zimbabwe	77,100,000	198,000,000	38.9	61.1
West	Nigeria, Niger, Benin, Togo	283,688,000	460,848,000	62.9	37.1

**Table 2 toxins-17-00493-t002:** Phenotype frequency data for CYP2B6, CYP2C9, CYP2C19, CYP2D6 and CYP3A4/5 in countries in Sub-Saharan Africa (SSA). Countries belonging to the same region (Central, East, South and West) are shown in the same colour.

CYP450	Country	PM Frequency	EM Frequency	N	Reference
CYP2B6	Ethiopia	0.090	0.910	264	[25]
	Niger Delta Ethnic population	0.440	0.560	50	[26] *
	Nigeria/Benin/Togo	0.180	0.820	226	[27]
	South Africa	0.130	0.870	81	[28]
	South Africa	0.220	0.780	80	[29]
	South Africa	0.120	0.880	163	[27]
	South Africa	0.230	0.770	60	[30]
	Sub Saharan Africa	0.675	0.325	961	[31] *
	Tanzania	0.150	0.850	153	[28]
	Tanzania	0.190	0.810	183	[25]
	Zimbabwe	0.140	0.860	100	[28]
	Zimbabwe	0.734	0.266	522	[32] *
CYP2C9	Ethiopia	0.043	0.957	69	[33]
	South Africa	0.025	0.975	100	[34]
	Zimbabwe	0.170	0.830	522	[32] *
CYP2C19	Nigeria	0.048	0.952	126	[35]
	South Africa	0.382	0.618	76	[36]
	South Africa	0.520	0.480	100	[37] *
	South Africa	0.480	0.520	75	[37] *
	South Africa	0.230	0.770	993	[38] *
	Tanzania	0.321	0.679	106	[36]
	Uganda	0.020	0.980	99	[39] *
	Zimbabwe	0.226	0.774	84	[36]
	Zimbabwe	0.048	0.952	84	[40]
CYP2D6	Burundi	0.050	n.d.	100	[41]
	South Africa	0.057	n.d.	98	[42]
	South Africa	0.592	0.408	76	[36]
	Tanzania	0.462	0.538	106	[36]
	Tanzania	0.070	n.d.	106	[43]
	Zimbabwe	0.037	0.940	103	[40]
	Zimbabwe	0.554	0.446	114	[36]
CYP3A4/5	Zimbabwe	0.726	0.274	522	[32]

* intermediate metabolizers were added to the poor metabolizers group; ultrarapid metabolizers were added to the extensive metabolizers group. PM = poor metabolizer; EM = extensive metabolizer; n.d. = not determined.

**Table 3 toxins-17-00493-t003:** Phenotype data per African region for CYP2B6, CYP2C9, CYP2C19, CYP2D6 and CYP3A4/5.

Region	CYP2B6		CYP2C9		CYP2C19		CYP2D6		CYP3A4/5	
	PM	EM	PM	EM	PM	EM	PM	EM	PM	EM
Central	0.150 *	0.850 *	0.024 *	0.976 *	0.038 *	0.962 *	0.050	0.950	0.180 *	0.820 *
East	0.136	0.864	0.043	0.957	0.176	0.824	0.266	0.734	0.180 *	0.820 *
South	0.456	0.544	0.147	0.853	0.261	0.739	0.301	0.699	0.726	0.274
West	0.227	0.773	0.024 *	0.976 *	0.048	0.952	0.031 *	0.969 *	0.180 *	0.820 *
SSA	0.440	0.560	0.136	0.864	0.235	0.764	0.255	0.745	0.726	0.274

* If no additional literature data was available, the data from *SouthAfrican_Population FW_Custom*, provided by SimCYP, was used.

**Table 4 toxins-17-00493-t004:** Observed and predicted mean pharmacokinetic parameters ± standard deviation for Central, East, South and West Africa.

	CENTRAL (dihydroartemisinin)		
	Observed [45]	Predicted	Predicted/Observed
C_max_ (mg/L)	0.812 ± 1.07	0.825 ± 0.370	1.02
T_max_ (h)	1.50 *	0.850 ± 0.130	0.570
AUC_ss_ (mg/L × h)	1.76 ± 1.86	1.25 ± 0.670	0.710
	**EAST (lamivudine)**		
	Observed [46]	Predicted	
C_max_ (mg/L)	1.10 ± 0.500	1.60 ± 0.610	1.45
T_max_ (h)	1.10 ± 0.800	1.27 ± 0.460	1.15
AUC_ss_ (mg/L × h)	5.60 ± 2.50	6.14 ± 1.89	1.10
	**SOUTH (ritonavir)**		
	Observed [47]	Predicted	
C_max_ (mg/L)	13.7 ± 3.04	7.13 ± 6.76	0.520
T_max_ (h)	4.00 ± 1.48	3.04 ± 0.620	0.760
AUC_ss_ (mg/L × h)	123 ± 36.7	71.2 ± 79.9	0.580
	**WEST (nifedipine)**		
	Observed [48]	Predicted	
C_max_ (mg/L)	0.205 ± 0.149	0.194 ± 72.4	0.950
T_max_ (h)	0.75 ± 2.59	0.300 ± 0.06	0.400
AUC_ss_ (mg/L × h)	0.605 ± 0.155	0.348 ± 0.202	0.58

* median value. The values in red do not meet the within twofold requirements for model verification.

## Data Availability

The original contributions presented in this study are included in the article/Appendix A. Further inquiries can be directed to the corresponding author(s).

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
