# Peer review of "Building Sub-Saharan African PBPK Populations Reveals Critical Data Gaps: A Case Study on Aflatoxin B1"

_toxins, 2025, doi:10.3390/toxins17100493_

Round 1

Reviewer 1 Report

Comments and Suggestions for Authors

This study analyzed the differences in CYP450 enzyme phenotype frequency among the four regions of sub Saharan Africa (SSA) - central, eastern, southern, and western, and constructed a region specific PBPK model. It was found that there were significant regional differences in CYP2B6, CYP2C19, and CYP2D6, confirming that SSA is not suitable for modeling as a single population and that more region specific data is needed to improve model accuracy.

  1. It is explicitly mentioned in the article that there is a serious lack of data coverage in various sub regions of sub Saharan Africa (such as only 1/9 of countries in the central region and only 3/14 of countries in the eastern region have data), which directly affects the universality of the conclusion. It is necessary to supplement the proportion of countries in each region that were not included in the study and their population weights.
  2. Missing key CYP450 enzyme data: This enzyme is crucial for AFB1 metabolism. The scarcity of CYP3A4 data affects the reliability of inter regional comparisons. Suggest re mining the CYP1A2/CYP3A4 allele frequencies in the African genome database. If there is insufficient direct data, cite the latest research literature.
  3. The discussion mentioned "data gaps" and "insufficient regional segmentation", but did not propose feasible improvement paths. It is recommended to use a mixed effects model to integrate existing data and digitize the contribution ratio of "intra country differences" and "inter regional differences".
  4. There are relatively few recent cutting-edge literature citations, so it is recommended to supplement more literature from the past five years.

5.Why are all regions based on the South African model? Can African Americans be added as an additional sample?

Reviewer 2 Report

Comments and Suggestions for Authors

This is a well-thought work. Nonetheless, some major revisions need to be addressed as described in the next comments.

Comment 1: Comment 1: Introduction should have examples of PBPK models applied to mycotoxins or other contaminants' exposure. Also, a final paragraph stating the importance of this work should be clearly given.

Comment 2: It is not clear why exclusion was based on the title individually, with 18 articles being excluded, since not always it is stated either in title or even in the abstract the specific population. This step should be revised.

Comment 3: It should be justified why only CYP450 phenotype data was collected.

Comment 4: Please provide the details of the software used (e.g. version used).

Comment 5: From the results section, Materials and Methods need to revise. For instance, lines 177 to 179 - calculation of frequencies; in line 189, it is stated the use of R studio (should be included in Materials and Methods, perhaps, a subsection with "Statistical analysis"). It is also stated that a systematic review was performed. How? What was the process? Was there software used? Rayyan?

Comment 4: Please provide the details of the software used (e.g. version used).

This is a good work even if the goal is to point out the limitations regarding the possibility of developing a PBPK model for AFB1 simulation. But, since the conclusion to this specific simulation was not possible due to the lack of data for the specific enzymes related to AFB1, then the authors should rethink the title towards this specific application. Along the manuscript it can be stated that if this would be an outcome, it would be needed more data. But there is data presented, where work was done and results obtained for other phenotypes. And this should be the aim. What you have achieve, and possible outcomes from this. In this matter, title should be revised, tables on supplementary data should be added as table within the text for better discussion.

Reviewer 3 Report

Comments and Suggestions for Authors

This study, titled "Unlocking Diversity: Building a PBPK population focusing on Sub-Saharan Africa for the simulation of AFB1 exposure," integrated regional data on CYP450 enzyme phenotypes into physiology-based pharmacokinetic (PBPK) models to assess the applicability of representing Sub-Saharan Africa (SSA) as a homogeneous population in simulating aflatoxin B1 (AFB1) exposure. The results revealed significant differences in CYP2B6, CYP2C19, and CYP2D6 phenotypes between regions, demonstrating that modeling SSA as a single group can lead to incorrect predictions of drug metabolism. It concludes that regional or even national subdivisions are necessary for greater accuracy, although the scarcity of in vivo data limits full validation of the models.

The title is clear and concise and reflects the focus of the manuscript. The abstract is self-contained and summarizes the objectives, methods, results, and conclusions; it is well-structured, without citations. The introduction contextualizes the relevance of PBPK models and justifies the need for population subdivision in SSA. Regarding the methodology, the authors describe the literature search, inclusion/exclusion criteria, and population construction in SimCYP, which is relatively clear, using recognized tools (EMA, FDA guidelines). In general, the results are adequate, using statistical tests (proportions with Bonferroni correction, Cohen's d). During the discussion, the authors acknowledge limitations (e.g., use of only PM/EM; scarcity of data; absence of CYP1A2), and the conclusions are consistent with the data presented: they recommend regional/national subdivision and highlight information gaps.

However, there appear to be some flaws in the reasoning, which are itemized for easier identification and verification.

#1_Inconsistency with AFB1 target: AFB1 metabolism is strongly dependent on CYP1A2 and CYP3A4 (lines 246–247, 340–343), but the study focused on CYP2B6, 2C9, 2C19, 2D6, and 3A4/5. The lack of data for CYP1A2 and the limitation for CYP3A4 compromise the direct link between modeling and AFB1 exposure. Therefore, the conclusion regarding relevance for AFB1 is weakened.

#2_Insufficient validation: For some regions, validation used pediatric data (lines 135–139, 309–310) or failed to meet the criterion within two (lines 220–221, 312–314). Despite this, the conclusion (lines 336–347) states that the models were “successfully verified (except for West Africa),” which is an exaggeration given the observed deviations.

#3_Generalization of statistical differences: Although phenotypic differences are significant (lines 187–197), the effects (Cohen’s d = 0.4–0.5) are only moderate (lines 316–317). The discussion could better relativize clinical relevance, rather than suggesting widespread impact (lines 242–249). #4_Methodology in the scientific literature: The systematic search is described (lines 86–91), but the protocol is not detailed (e.g., specific exclusion criteria, risk of bias, study quality assessment). Methodological transparency is limited, making it difficult.

#5_Figures and Tables: Some tables (e.g., Tables 1–2, lines 170–185) are clear, but the main graphs (S1–S5) are in supplements and not described in detail in the text. This reduces the clarity of the presentation of the main results.

As a final suggestion to the authors, it can be said that it is essential to ensure the reproducibility of the study. The methodology is acceptable, but would need more detail to fully meet reproducibility standards.

Reviewer 4 Report

Comments and Suggestions for Authors

This research article deals with an investigation of the prevalence of clinical CYP450 phenotypes across Sub-Saharan Africa (SSA) to determine the feasibility of defining SSA as a single population. This is an interesting piece of work, however, the following points need to be addressed for a possible publication in Toxins:

  1. The abstract need to be enriched with some key quantitative data. Also include the detail of approximate number of studies used for each region.
  2. “Sub-Saharan Africa” should be included as one of the keywords.
  3. L62-69: although the text here discusses Africa’s genetic diversity, the information on why these matters for PBBK especially by linking high polymorphism rates to modeling challenges.
  4. L99-101: the use of ‘SouthAfrican_Population FW_Custom’ as the baseline is justified, However, the rationale should be stronger by clarifying why this population was chosen instead of building de novo models.
  5. L116-131: the description on missing data handling is not clear and thus needs to clarified on how gaps in phenotype frequency were addressed.
  6. L123-126: Although doses of probe drugs are provided, there is no justification on why these specific doses were chosen.
  7. A new section ‘statistical analyses’ should be included detailing the statistical method and software used in this study for data analysis.
  8. L166-169: this template note should be removed.
  9. Tables 1 & 2: there are extensive numbers and mixed data types, which are difficult to interpret. Use consistent decimal formatting (probably 2 decimals throughout).
  10. L184-185: the footnote information “If no … used (*)” should be moved to the footnotes.
  11. L187-197: the reported Cohen’s d values (0.4-0.5) should be biologically interpreted through discussing potential clinical implications of these moderate differences.
  12. L202-209: the details of ‘SouthAfrican_Population FW_Custom’ given here should be moved to ‘Methods’ section.
  13. Table 3: Why some predicted PK values fall outside the “within twofold” acceptance range? Please explain. Also discuss on why model performance varied across regions.
  14. L256-266: the discussion about Figure 3 should be moved earlier to emphasize incomplete datasets.
  15. L279-285: Discuss how the limitation of CYP3A4/5 data scarcity significantly impacts the model accuracy.
  16. L316-319: the Cohen’s d interpretation should include potential clinical consequences especially for narrow therapeutic index drugs.
  17. L338-341: the authors can simply state the AFB1 metabolism modeling remains preliminary.
  18. L348-350: A brief statement on potential strategies such as pan-African biobanks or collaborative databases should be included.
  19. References: inconsistent formatting of references should be double-checked.
  20. Figures S1-S5: Key findings of these figures should be summarized in the main text substantially.

Round 2

Reviewer 1 Report

Comments and Suggestions for Authors

NONE

Reviewer 4 Report

Comments and Suggestions for Authors

The authors have satisfactorily addressed all the comments raised by reviewers and substantially improved the overall quality of the article. Therefore, I recommend accepting this article for publication in Toxins.